# ADVERSARIAL BOTTLENECK METHOD FOR VISION-LANGUAGE LARGE MODEL EXPLAINABILITY

## ABSTRACT

Nowadays CLIP is a leading vision-language model, showing strong functionality, especially in tasks like search engine matching. However, its high performance is often accompanied by the complexity of the decision-making process, making the interpretability of the model a major challenge. Existing XAI methods mainly focus on unimodal settings, with state-of-the-art methods often being attribution algorithms based on adversarial attacks. These methods perform well in unimodal tasks such as image classification. However, expanding these methods to handle cross-modal tasks (such as image-text alignment and cross-modal retrieval) presents several obstacles. For multimodal tasks, the most effective XAI methods currently rely on the bottleneck principle, which limits information flow to analyze model decisions. In this paper, we propose a new approach that integrates adversarial attribution methods with the bottleneck principle. This approach not only interprets multimodal models such as CLIP but also preserves the advantage of unimodal attribution algorithms in precisely identifying key features that influence model decisions within a specific modality. By introducing our model, we can obtain a more robust and broadly applicable representation for vision-language models, further enhancing their transparency and trustworthiness in complex tasks. Comprehensive experiments demonstrate that, compared to state-of-the-art XAI methods, our approach improves the interpretability of text and images by 69.12% and 19.36%, respectively. Our code is available at `https://anonymous.4open.science/r/ABM-5C28/`.

## 1 INTRODUCTION

Nowadays large-scale Vision-Language Models (VLMs) have gained significant attention for their outstanding performance in various multimodal tasks (Wasim et al., 2023), including image-text matching (Peng et al., 2023), zero-shot classification (Novack et al., 2023), and image retrieval (Baldrati et al., 2022; Sain et al., 2023). CLIP (Radford et al., 2021), developed by OpenAI [1], learns joint representations from a large number of image-text pairs, making it highly effective in tasks requiring understanding of visual and textual data. As VLMs are increasingly integrated into various applications, their reliability and interpretability are becoming increasingly important. Therefore, Explainable Artificial Intelligence (XAI) has emerged as a crucial component in understanding and interpreting the decision-making processes of VLMs. XAI not only helps build trust in AI systems, but also plays a key role in ensuring transparency (Humer & Strobelt, 2023), especially when these models are used in sensitive applications like web services where erroneous or biased decisions can have significant real-world consequences. For multimodal models such as CLIP, there is a growing demand for robust XAI methods, as it is essential to scrutinize whether the model's explanations for image-text pairs are free from bias, fair, and accurate.

Although various XAI methods designed for unimodal classification tasks have matured (Selvaraju et al., 2017; Petsiuk et al., 2018; Zhu et al., 2024b), directly modifying their loss functions for VLM tasks always leads to a significant drop in performance. Moreover, current interpretability algorithms designed for VLM tasks require substantial sampling, which introduces information interference in explaining a single sample (Kalibhat et al., 2023; Lei et al., 2024; Gandelsman et al., 2023). Due to the interpretability variations caused by different sampling outcomes, it is challenging to determine

---

[1] `https://openai.com/`

whether the explanation originates from the sample itself or the generated samples. Since a single sample pair can already generate a relevance score (i.e., the degree of image-text matching) and the VLMs understand the semantic information in both text and image, there must be an approach capable of interpreting a single sample pair without relying on extensive sampling.

M2IB (Wang et al., 2023) is one of the few algorithms designed to interpret a single sample pair in VLMs. It leverages the Information Bottleneck Principle (IBP) (Tishby et al., 2000) but inevitably introduces too much human intuition, leading to interference in the explanation itself. This makes it difficult to trust whether the explanation comes from the model or human-imposed rules, causing a crisis of trustworthy in the interpretability algorithm. Additionally, M2IB uses gradient descent as the optimization strategy for iterating over the model parameter design. Our findings show that the parameters do not fully achieve the desired Information Bottleneck compression during the optimization process.

In unimodal tasks, the state-of-the-art XAI methods are adversarial attribution methods. We summarized their common principles and extracted the Adversarial Attribution Theory (AAT). While AAT also relies on human intuition, it provides an insight into exploring the Information Bottleneck in IBP—specifically, by conducting adversarial attacks on the bottleneck. Based on this insight, we proposed a novel approach called Adversarial Bottleneck Method (ABM) to eliminate the issues of extensive human intuition and hyperparameter tuning. By using hyperparameters tuned solely for accuracy — where higher values lead to greater precision — our ABM achieves significant performance improvements. Our contributions are as follows:

- We identify critical limitations when applying the Information Bottleneck Principle (IBP) and Adversarial Attribution Theory (AAT) to vision-language models, particularly in cross-modal interpretability tasks.
- We propose the Adversarial Bottleneck Method (ABM), a novel framework that fuses adversarial attribution with the bottleneck principle to enhance cross-modal interpretability while minimizing reliance on heuristic design and extensive hyperparameter tuning. We also provide rigorous theoretical foundations for ABM.
- We demonstrate that ABM significantly improves interpretability across modalities, achieving relative gains of 69.12% for text and 19.36% for images over state-of-the-art XAI methods. Our approach is robust, generalizable, and supported by a publicly available open-source implementation.

## 2 RELATED WORK

### 2.1 TRADITIONAL INTERPRETABILITY METHODS

With the rise of multimodal models like CLIP, Explainable AI faces new challenges in interpreting their decision-making processes. Traditional interpretability methods, mainly developed for unimodal tasks, include both model-specific techniques and model-agnostic "black-box" approaches. Gradient-based methods, such as Saliency Maps (Simonyan, 2013) and Grad-CAM (Selvaraju et al., 2017), offer intuitive explanations by analyzing gradients but are typically limited to unimodal tasks and not easily extendable to VLM tasks. Model-agnostic methods like RISE (Petsiuk et al., 2018) and LIME (Ribeiro et al., 2016) have emerged as more scalable solutions for VLMs. RISE generates global explanations by occluding inputs, but random sampling can neglect important features. LIME provides local explanations by approximating decisions with linear models, but local approximations can lead to inconsistent results. Overcoming the limitations of these methods, particularly in feature accuracy and consistency, is important. The Integrated Gradients (IG) framework (Sundararajan et al., 2017) introduces two key concepts—the Sensitivity Axiom and the Implementation Invariance Axiom—which ensure that explanations accurately reflect feature influence and are independent of model implementation details.

### 2.2 ADVERSARIAL ATTRIBUTION METHODS

Since the IG method requires a well-defined baseline, it is challenging to determine an appropriate baseline in practice. Advanced interpretability algorithms currently use adversarial attacks to find adaptive baselines. AGI (Pan et al., 2021) is an early attribution method that applied adversarial

attacks, targeting highly confident classes in classification tasks. However, this cannot be directly extended to VLM tasks. The More Faithful and Accelerated Boundary-Based Attribution (MFABA) method (Zhu et al., 2024b) significantly improves computational efficiency by introducing the second-order Taylor expansion and multi-step gradient ascent, while providing more precise and robust explanations, further addressing AGI's shortcomings. In our experiments, we will compare the performance of MFABA when its loss function is modified for VLM tasks. AttEXplore (Zhu et al., 2024a) introduces non-linear integrated paths and frequency domain information through transferable adversarial attacks, enhancing the exploration of model parameters and reducing reliance on fixed adversarial paths. This allows it to cross multiple decision boundaries, improving the accuracy and robustness of attribution results. Compared to other methods, AttEXplore not only offers significant improvements in computational efficiency, but also achieves broader generalization by incorporating transferable adversarial attack techniques. However, since AttEXplore requires class information for attribution tasks, it is not suitable for CLIP.

### 2.3 CLIP INTERPRETABILITY METHODS

Many methods for multimodal interpretability, including COCOA (Lin et al., 2022), TEXTSPAN (Gandelsman et al., 2023), and LICO (Lei et al., 2024), introduce external information during the attribution process, which can compromise fairness and transparency. For instance, COCOA adjusts IG's loss function and incorporates positive and negative samples, but its reliance on sampling hinders direct explanation of a given sample. Similarly, TEXTSPAN needs a predefined text set, limiting its generalization, while LICO retrains a new model for attribution, introducing randomness in the sampling process. These methods often rely on external factors, which makes it difficult to exclude extraneous influences. The core issue is that attribution should enhance model transparency without introducing external information that could affect the results.

Chefer et al. (2021) proposed a Transformer-based interpretability method, generating relevance maps using attention weights, but it relies heavily on model structure and lacks an information-theoretic framework for quantifying attribution results. Building on gradient-based explanations, Zhao et al. (2024) introduce Grad-ECLIP, a CLIP-specific method that decomposes the transformer encoders and relates the image–text matching score to intermediate token features, producing fine-grained heatmaps over both image regions and textual tokens. However, Grad-ECLIP still operates purely in the gradient space and does not provide an information-theoretic characterization of what information is preserved or discarded in CLIP's representations.

Tishby et al. (2000) introduced the Information Bottleneck Method, which retains only relevant information by compressing irrelevant parts of the input signal. Schulz et al. (2020)'s extension of this theory applies noise in intermediate layers to quantify effective information, enabling more precise feature attribution than traditional gradient-based methods. Wang et al. (2023) introduced M2IB, a multimodal interpretability method based on the Information Bottleneck Theory that maximizes mutual information between images and text to enhance the interpretability. M2IB does not rely on task-specific labels, making it suitable for unlabeled data, and helps analyze the complex relationships between image and text inputs. Besides, adversarial attribution methods, although successful in unimodal tasks, have not yet been applied to multimodal models like CLIP. This paper combines adversarial attacks with multimodal interpretability to further improve the understanding of such models.

## 3 METHODS

### 3.1 PRELIMINARIES

We follow the established setup of CLIP (Radford et al., 2021), which employs a pre-trained image-text representation model. This model consists of two main encoders: the image encoder $f_{\mathcal{I}} : \mathbb{R}^n \to \mathbb{R}^k$, which converts the input image $x_{\mathcal{I}} \in \mathbb{R}^n$ into a $k$-dimensional image representation, and the text encoder $f_{\mathcal{T}} : \mathbb{R}^m \to \mathbb{R}^k$, which converts the input text $x_{\mathcal{T}} \in \mathbb{R}^m$ into a $k$-dimensional text representation. The similarity between the visual and textual modalities can be measured using the cosine similarity $\cos \langle f_{\mathcal{I}}(x_{\mathcal{I}}), f_{\mathcal{T}}(x_{\mathcal{T}}) \rangle$, and these representations can also be used for downstream tasks such as classification and retrieval. In the following sections, we refer to the encoder as $f$, representing either $f_{\mathcal{I}}$ or $f_{\mathcal{T}}$ (with $f_{\mathcal{I}}$ for the image modality and $f_{\mathcal{T}}$ for the text modality). For a

neural network with $L$ layers, we decompose the output $f(x)$ into two consecutive sub-networks: $f^{l-L}(x)$, representing the transformation from the intermediate layer $l$ to the last layer $L$, and $f^{1-l}(x)$, representing the transformation from the input layer to the intermediate layer $l$. The composition $f(x)$ is then expressed as $f^{1-l} \circ f^{l-L}(x)$, where $f^{1-l}(x)$ is applied first, followed by $f^{l-L}$. For ease of interpretation, we use $z = f^{1-l}(x)$ to denote the latent features at the intermediate layer $l$. The goal of our work is to construct an explanation method $A$ that provides a result $A(x) \in \mathbb{R}^{|x|}$. The higher the value of $A(x)$, the more important the corresponding dimension of the representation, highlighting the influence of different parts of the input. In this case, the features of the intermediate layer $z$ play a crucial role in understanding and interpreting the model's behavior. In this paper, we use $\tilde{z}$ to denote the distribution of the intermediate layer features that we set.

## 3.2 ADVERSARIAL BOTTLENECK METHOD

In this section, we first introduce the theoretical foundations of the Information Bottleneck Principle (IBP) and The Adversarial Attribution Theory (AAT). We then identify the issues present in applying these two theories and provide a rigorous derivation that proves the feasibility of optimizing IBP using adversarial attacks, which we term the Adversarial Bottleneck Method (ABM). Most of our demonstrative derivations are placed in the Appendix, ensuring that the results are both valid and elegantly presented.

### 3.2.1 INFORMATION BOTTLENECK PRINCIPLE (IBP) AND ADVERSARIAL ATTRIBUTION THEORY (AAT)

IBP is derived from the concept of mutual information in information theory and is intuitively easy to understand. Therefore, various studies have applied IBP theory to XAI research (Tishby et al., 2000; Schulz et al., 2020; Wang et al., 2023). To our best knowledge, as discussed in related work (Section 2.3), M2IB (Wang et al., 2023) is among the few interpretability algorithms specifically designed for VLM tasks that operate on a *single* image–text pair without relying on additional samples or retraining. In contrast, classical attribution methods such as Integrated Gradients (Sundararajan et al., 2017) or Grad-CAM (Selvaraju et al., 2017) were originally developed for unimodal classifiers and must be adapted to the CLIP cosine-similarity objective and intermediate layers to yield usable attributions in the cross-modal setting. The core objective of M2IB is as follows:

$$\alpha_m^* = \max_{\alpha_m} I(\tilde{z}_m, e_{m'}; \alpha_m) \quad s.t. \quad I(\tilde{z}_m, x_m; \alpha_m) \leq \bar{I} \tag{1}$$

and then it can be transformed into:

$$\alpha_m^* = \max_{\alpha_m} I(\tilde{z}_m, e_{m'}; \alpha_m) - \beta I(\tilde{z}_m, x_m; \alpha_m) \tag{2}$$

where $\tilde{z}_m \in \mathbb{R}^k$ represents the encoding of the features in the modality $m$, $e_{m'}$ represents the encoding in modality $m'$ (where if $m$ denotes the vision modality, $m'$ represents the text modality, and vice versa), and $x_m$ denotes the input of the modality $m$. $\alpha_m$ is the parameter that controls the size of the bottleneck, and $\bar{I}$ is a compression constraint. **Intuitively, the idea of M2IB is to train $\alpha_m$ to maximize the mutual information between the encoding of one modality and the other while minimizing the mutual information between the encoding and the input of that modality.** Here is a brief supplement of the theory of mutual information (more details in (Wang et al., 2023)): the mutual information $I(X;Y)$ between events $XY$ can be expressed as the difference between two entropies (entropy measures the uncertainty of an event; the higher the entropy, the greater the uncertainty) $I(X;Y) = H(X) - H(X \mid Y)$, which means that observing the event $Y$ significantly reduces the uncertainty of the event $X$, indicating that the two are strongly related.

Equation 2 introduces $\beta$, a hyperparameter used to control the balance between the mutual information of the encoded representation and the original input. It shows that we seek an encoding that is as uncorrelated as possible with the original input while still retaining its function in VLMs. The retained information is the critical information that we aim to obtain as interpretability results. The latent feature is controlled using $\tilde{z}_m = \sigma(\alpha_m) \cdot z + (1 - \sigma(\alpha_m)) \cdot \varepsilon, \varepsilon \sim \mathcal{N}(0, 1)$, where $\sigma$ denotes the sigmoid function, which maps the values of the parameters between 0 and 1. As the corresponding dimension of $\sigma(\alpha_m)$ approaches 0, $\tilde{z}_m$ approaches independent noise, and the mutual information with the input decreases.

However, M2IB faces notable challenges in practical implementation, particularly due to the sensitivity and instability associated with tuning the hyperparameter $\beta$. We note that such parameter should

not be the same for each sample, model, or modality. We cannot predict which $\beta$ will balance the two mutual information terms. If the model uses more information for a given sample, then $\beta$ should be reduced to lower the importance of the second mutual information term and vice versa. Unfortunately, the choice of $\beta$ is often empirical rather than theoretically derived. The empirical results supporting this observation can be found in **Appendix F.5**. Additionally, the reduction of mutual information $I(\tilde{z}_m, x_m; \alpha_m)$ is limited, which is caused by the value of $\beta$ and the optimization strategy used for $\alpha_m$. In M2IB, gradient descent is used to optimize $\alpha_m$, and to ensure that the initial optimization retains as much $I(\tilde{z}_m, x_m; \alpha_m)$ as possible, all dimensions of $\alpha_m$ are set to relatively large values (in M2IB, this is set to 5, where $\sigma(5) = 0.9933$). However, during the update process, the gradient values of $\alpha_m$ are very low, we calculated the proportion of gradients falling between $-0.00005$ and $0.00005$, which accounted for 99.51%, 99.50%, and 99.48% in the Conceptual Captions (Sharma et al., 2018), ImageNet (Deng et al., 2009), and Flickr8k datasets (Hodosh et al., 2013), respectively, resulting in limited updates and incomplete exploration. These hyperparameters create a crisis of trust in attribution algorithms because modifying them produces vastly different results. How can we trust an interpretability algorithm that introduces uncertainty and human intuition, and what attribution result should we believe? Motivated by the theoretical foundations and practical success of adversarial attribution algorithms, we propose a reformulation of the Information Bottleneck Principle (IBP) that integrates their distinctive properties. Before that, current adversarial attribution algorithms can be summarized as Adversarial Attribution Theory (AAT), as shown in Equation 3 (Pan et al., 2021; Zhu et al., 2024b;a).

$$\sum_i \int \Delta z_i^t \odot g(z_i^t)\, \mathrm{d}t = L(z^T) - L(z^0) \tag{3}$$

Here $i$ denotes the $i$-th dimension, and $D$ represents the total number of dimensions. We denote $g(z^t) = \frac{\partial L(z^t)}{\partial z^t}$ as the gradient of the loss with respect to the latent representation at step $t$. The difference between the loss function $L(\cdot)$ at the final state $z^T$ and the initial state $z^0$ can thus be expressed as the accumulated contributions of each dimension along the path, with the $i$-th dimensional contribution written as $\int \Delta z_i^t \odot g(z_i^t)\, \mathrm{d}t$, where $\odot$ denotes element-wise multiplication. In other words, $\int \Delta z_i^t \odot g(z_i^t)\, \mathrm{d}t$ defines the attribution score assigned to latent dimension $i$, and summing these scores over all $i$ exactly recovers the total loss change $L(z^T) - L(z^0)$. Our aim here is not to introduce a new optimization theorem, but to adapt this standard path-integral view of gradient-based updates to the adversarial attribution setting and make the per-dimension decomposition explicit.

We use the first-order Taylor expansion to approximate the loss function along the optimization path:

$$L(z^t) = L(z^{t-1}) + \left( \frac{\partial L(z^{t-1})}{\partial z^{t-1}} \right)^{\top} (z^t - z^{t-1}) + \varepsilon$$

$$\sum_{t=1}^{T} L(z^t) \approx \sum_{t=0}^{T-1} L(z^t) + \sum_{t=0}^{T-1} \left( \frac{\partial L(z^t)}{\partial z^t} \right)^{\top} (z^{t+1} - z^t)$$

$$L(z^T) - L(z^0) \approx \sum_{t=0}^{T-1} \left( \frac{\partial L(z^t)}{\partial z^t} \right)^{\top} (z^{t+1} - z^t) \tag{4}$$

$$\approx \sum_{t=0}^{T-1} g(z^t)^{\top} \Delta z^t \ \approx \ \int \Delta z^{t\top} g(z^t)\, \mathrm{d}t \ \approx \ \sum_{i=1}^{D} \int \Delta z_i^t\, g_i(z^t)\, \mathrm{d}t$$

Here $\varepsilon$ represents the remainder term of the Taylor expansion, which is a higher-order infinitesimal. In the analysis of neural networks, it is a standard assumption to neglect such higher-order terms, since they vanish asymptotically and do not affect the first-order characterization of the optimization dynamics. We denote $\Delta z^t = z^{t+1} - z^t$ as the update at step $t$. Under this formulation, if the contribution term of a given dimension is always zero along all admissible paths, then changing that feature cannot affect the loss, which corresponds to the Sensitivity axiom in attribution methods. Meanwhile, summing the contributions over all dimensions recovers $L(z^T) - L(z^0)$, which corresponds to the Completeness axiom. Thus, this path-based adversarial attribution formulation satisfies Sensitivity and Completeness by construction, rather than imposing these properties heuristically. The central premise of AAT lies in leveraging adversarial perturbations—formulated in Equation 5—as the update term $\Delta z^t$, while simultaneously monitoring the contribution of each latent dimension throughout

the optimization process. In this sense, Equations equation 3 and equation 4 describe the general path-based attribution mechanism, and Equation 5 instantiates a specific projected gradient-ascent path (with an $\ell_\infty$-norm constraint) within this framework.

$$z^{t+1} = z^t + \Delta z^t = z^t + \eta \cdot \text{sign}\left(\frac{\partial L(z^t)}{\partial z^t}\right) \tag{5}$$

While AAT has demonstrated remarkable performance in classification tasks, its direct application to VLM tasks presents a range of practical challenges. The most obvious issue is the design of the loss function. For VLM tasks, cosine similarity is usually used as the result of modality correlation. However, unlike cross-entropy loss, cosine similarity is a periodic function, making it difficult to determine an appropriate learning rate. Furthermore, the cumulative process of calculating contributions in AAT may introduce redundancy. This limitation is non-trivial and warrants further investigation, as illustrated by the following example.

*Illustrative Example: A parameter has a value of $1$ in the first time step and its gradient is $-0.5$. With a fixed learning rate of $1$, after updating according to Equation 5, the value becomes $0$, and the cumulative contribution is $-1 \times -0.5 = 0.5$. In the second time step, the gradient is $0.5$, so the value returns to $1$, and the cumulative contribution is $0.5 + 1 \times 0.5 = 1$. After two updates, the parameter value remains unchanged, yet it has a contribution value of $1$, which is unreasonable. This redundancy is especially problematic when using AAT with functions like cosine similarity that frequently oscillate.*

The core issue lies in the fact that the M2IB algorithm uses the final state, but due to the difficulty in confirming the hyperparameters and the use of an unreasonable optimization strategy, it is difficult to trust. AAT updates parameters using adversarial attacks and calculates the cumulative state, but due to redundancy and difficulty in determining the learning rate—both of which are especially problematic in VLM—the results are unsatisfactory. Our ultimate goal is to construct a new theory that aligns with the objectives of the IBP while addressing these flaws, which we term the Adversarial Bottleneck Method (ABM).

### 3.2.2 REASONING FOR THE ADVERSARIAL BOTTLENECK METHOD (ABM)

Before introducing the specific steps of our algorithm, we present our core theory:

**Theorem 3.1** (Main property of ABM). *Given a constraint method $C_z$ and an update rule*

$$z^{t+1} = C_z\left(z^t + \frac{z}{T} \cdot \text{sign}\left(\frac{\partial I(\tilde{z}^t, e_{m'})}{\partial z^t}\right)\right), \tag{6}$$

*when $T$ is sufficiently large (so that the step size $\frac{z}{T}$ is small and the first-order Taylor approximation of $I(\tilde{z}^{t+1}, e_{m'})$ is valid), the following conditions hold:*

$$I(\tilde{z}^{t+1}, x_m) \leq I(\tilde{z}^0, x_m), \tag{7}$$

$$I(\tilde{z}^{t+1}, e_{m'}) > I(\tilde{z}^t, e_{m'}). \tag{8}$$

Here $T$ represents the number of update steps and $z^0 = z$. The vector $z$ denotes the latent representation, and the ratio $\frac{z^t}{z}$ is taken element-wise.

**Clarifications.** The constraint method $C_z(x_i)$ is defined as:

$$C_z(x_i) = \begin{cases} \max\left(\min(x_i, 0), z_i\right), & \text{if } z_i < 0, \\ \min\left(\max(x_i, 0), z_i\right), & \text{if } z_i \geq 0. \end{cases} \tag{9}$$

**Motivation of the constraint.** The motivation for introducing $C_z$ is to prevent the updates of each latent dimension from drifting outside their feasible range. In practice, adversarial updates without constraints can cause $z^t$ to overshoot, leading to degenerate or unstable mutual information estimates. By bounding $z^t$ between its original value and zero (depending on the sign of $z_i$), the constraint ensures that each dimension remains within a meaningful interval. This has two important effects: (i) it guarantees $\frac{z^t}{z} \in (0, 1)$ so that the interpolation in Equation 10 is well-defined, and (ii) it stabilizes the optimization so that the inequalities in Theorem 3.1 (Eq. 7 and Eq. 8) hold consistently.

The update of $\tilde{z}^t$ is given by

$$\tilde{z}^t = \frac{z^t}{z} \cdot z + \left(1 - \frac{z^t}{z}\right) \cdot \varepsilon = z^t + \left(1 - \frac{z^t}{z}\right) \cdot \varepsilon, \quad \varepsilon \sim \mathcal{N}(0, I). \tag{10}$$

Equivalently, letting $\gamma_t = \frac{z^t}{z} \in [0, 1]$ (element-wise), we can rewrite Equation 10 as

$$\tilde{z}^t = \gamma_t z + (1 - \gamma_t)\varepsilon, \tag{11}$$

so that $\gamma_t$ acts as a gate controlling how much of the original information in $z$ is retained versus replaced by noise.

Equation 10 thus constructs the perturbed representation $\tilde{z}^t$ as a *controllable mixture* of the original latent $z$ and Gaussian noise $\varepsilon$, modulating at each iteration how much input information is retained. In the two limits, $z^t \to 0 \Rightarrow \tilde{z}^t \to \varepsilon$ (pure noise, no input information), and $z^t \to z \Rightarrow \tilde{z}^t \to z$ (full retention of the original information). This formulation enables a continuous, controlled compression of mutual information with the input $x$. The update rule in Theorem 3.1 uses the gradient of $I(\tilde{z}^t, e_{m'})$ with respect to $z^t$ to guide $z^t$ along this interpolation path, while $C_z$ keeps $z^t$ within $[0, z]$. The proof leverages this construction to show that, under the stated dynamics, the conditions of Theorem 3.1 hold, thereby aligning the update rule with Eq. 10. Importantly, Equation 10 is an Information-Bottleneck–motivated parametrization that maps the updated gate $z^t$ to the actual bottleneck representation $\tilde{z}^t$, rather than a closed-form expression derived directly from mutual information.

We adopt the cosine similarity design from M2IB (Wang et al., 2023) for $I(\tilde{z}^t, e_{m'})$. For simplicity, we omit the subscript $m$ in $\tilde{z}$ in Theorem 3.1. The detailed proof of Theorem 3.1 is provided in **Appendix A**.

The significance of Theorem 3.1 lies in discovering an update method that aligns with the original IBP definition, fulfilling the objective in Equation 1, where the compression constraint $\bar{I}$ is defined as $I(\tilde{z}^0, x_m)$. Through iterative updates, we can find the optimal $\tilde{z}_i^t$ that maximizes $I(\tilde{z}^t, e_{m'})$. Since we set the adversarial attack learning rate as $\frac{z}{T}$, we can fully explore the process from $z$ to $\varepsilon$. For example, if a dimension $z$ is not important for $\partial I(\tilde{z}^t, e_{m'})$, we have $\text{sign}\left(\frac{\partial I(\tilde{z}^t, e_{m'})}{\partial z_i^t}\right) = -1$ at each time step. After $T$ updates, the updated dimension becomes $z_i + (-1) \cdot T \cdot \frac{z_i}{T} = 0$. Then we can use $I(\tilde{z}_i^t, x)$ as the importance of dimension $i$. In practical computation, we have:

$$A(z_i) = KL\big(P(\tilde{z}_i^T \mid x) \,\|\, \mathcal{N}(0, 1)\big) = \frac{1}{2}\left(-1 - \log\left(\left(1 - \frac{z_i^T}{z}\right)^2\right) + (z_i^T)^2 + \left(1 - \frac{z_i^T}{z}\right)^2\right). \tag{12}$$

Since the feature space information has not changed, we adopt linear interpolation similar to Grad-CAM to obtain the interpretability results of the original feature dimension. Detailed information on how Equation 12 is obtained can be found in **Appendix B**. From the iterative process, it is clear that, while adhering to Equation 1, we do not use M2IB's hyperparameter $\beta$ to balance the two mutual information terms, and we fully explore the process from $z$ to $\varepsilon$. Moreover, by eschewing the accumulation of intermediate states, our approach circumvents the redundancy and learning rate sensitivity issues inherent in AAT. For implementation clarity, the complete pseudocode of ABM is given in Appendix C.

## 4 EXPERIMENTS

### 4.1 EXPERIMENTAL SETUP

We evaluate on ViT-B/32 (Dosovitskiy, 2020) for multimodal interpretability. ViT-B/32 processes images as patch sequences via a Transformer, offering strong global context modeling. Experiments span three representative datasets: Conceptual Captions (Sharma et al., 2018), ImageNet (Deng et al., 2009), and Flickr8k (Hodosh et al., 2013).

Baselines include M2IB (Wang et al., 2023), RISE (Petsiuk et al., 2018), Grad-ECLIP (Zhao et al., 2024), Grad-CAM (Selvaraju et al., 2017), Chefer et al. (Chefer et al., 2021), Saliency Map (Simonyan, 2013), MFABA (Zhu et al., 2024b), and FastIG (Hesse et al., 2021). Unless otherwise noted, we follow each paper's default settings for fair comparison. Our ABM uses target layer 9 and $T{=}10$ iterations. All experiments are run on Linux with CUDA 12.4 and two NVIDIA A100 GPUs.

## 4.2 EVALUATION METRICS

In this experiment, we follow the evaluation protocol used in M2IB (Wang et al., 2023), utilizing **Confidence Drop** and **Confidence Increase** (Chattopadhay et al., 2018) as primary metrics to measure the performance of attribution methods. These metrics quantify how much the model's confidence changes when critical features identified by an attribution method are removed (Confidence Drop) or when noncritical features are removed (Confidence Increase), thereby providing insights into the reliability of the attribution methods in terms of faithfulness to the model's decision process. Specifically, a lower **Confidence Drop** indicates better interpretability, as it suggests that the remained features are indeed crucial to the model's decision. Conversely, a higher **Confidence Increase** signifies better interpretability, as it implies that the removal of less important features reduces noise and enhances the model's confidence. For text modality evaluation, following M2IB's setup, we use a Boolean-based criterion, ensuring that the evaluation metrics remain invariant across different settings. Details are provided in **Appendix D**. We also report evaluation results under the **ROAD** framework (Rong et al., 2022), which provides a consistent and retraining-free assessment of attribution faithfulness by mitigating information leakage during feature removal.

## 4.3 EXPERIMENTAL RESULTS

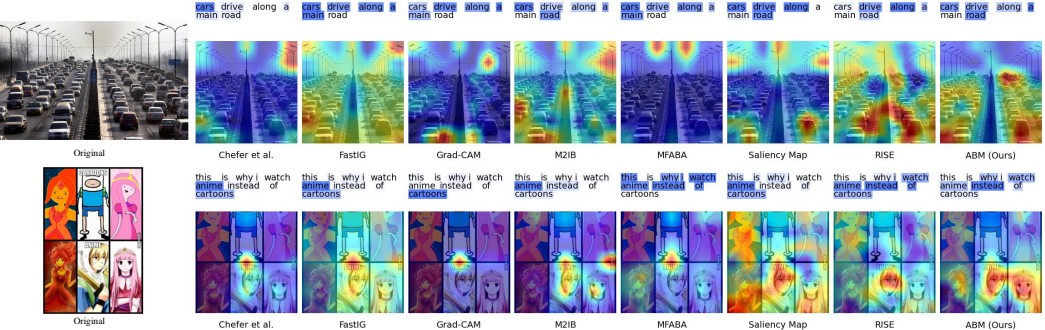

Figure 1: Interpretability comparison between our ABM method and other baseline methods on Conceptual Captions dataset.

Table 1: Experimental results comparing our ABM method with various baseline methods on three datasets. **Confidence Drop** ↓ indicates lower values are better, and **Confidence Increase** ↑ indicates higher values are better.

| Method | Conceptual Captions | | | | ImageNet | | | | Flickr8k | | | |
|---|---|---|---|---|---|---|---|---|---|---|---|---|
| | Image | | Text | | Image | | Text | | Image | | Text | |
| | Conf Drop ↓ | Conf Incr ↑ | Conf Drop ↓ | Conf Incr ↑ | Conf Drop ↓ | Conf Incr ↑ | Conf Drop ↓ | Conf Incr ↑ | Conf Drop ↓ | Conf Incr ↑ | Conf Drop ↓ | Conf Incr ↑ |
| M2IB | 1.1171 | 39.3 | 1.706 | 37.4 | 1.1615 | 49.4 | 2.6018 | 25.4 | 1.4731 | **28.1** | 2.0783 | 34.7 |
| RISE | 1.4197 | 28.8 | 0.8002 | 43.95 | 1.001 | 54 | 0.9928 | 46.8 | 0.8914 | 5.7 | 2.1114 | 46.4 |
| Grad-ECLIP | 2.3956 | 26.5 | 1.2894 | 43.2 | 2.3199 | 29.2 | 2.0104 | 28.3 | 7.7444 | 2.6 | 2.1114 | 37.5 |
| Grad-CAM | 4.1064 | 20.2 | 1.7994 | 34.4 | 2.5483 | 33.9 | 2.6424 | 25.7 | 5.1869 | 13.6 | 2.1823 | 34.2 |
| Chefer et al. | 2.0138 | 33.65 | 0.9333 | 45.3 | 1.6636 | 44 | 1.6732 | 29.9 | 2.6214 | 26.8 | 1.362 | 42.6 |
| Saliency Map | 10.4351 | 2.95 | 1.0723 | 40.05 | 4.7331 | 16.4 | 1.7631 | 33.1 | 12.154 | 0.1 | 1.0797 | 45.9 |
| MFABA | 10.1878 | 2.6 | 1.0503 | 36.25 | 5.0242 | 12.7 | 1.7437 | 28.5 | 12.07 | 0.1 | 1.1551 | 42.6 |
| FastIG | 10.5117 | 2.9 | 0.9718 | 41.25 | 4.7905 | 16.9 | 1.6486 | 34.8 | 12.2244 | 0.1 | 1.3098 | 43.9 |
| ABM | **0.7878** | **43** | **0.005019** | **44.5** | **0.746** | **57.1** | **0.0049** | **60.4** | **1.1169** | 26.8 | **0.0039** | **59.3** |

Table 1 presents the quantitative results across three datasets: Conceptual Captions, ImageNet, and Flickr8k. Our ABM method consistently demonstrates superior performance in both **Confidence Drop** and **Confidence Increase** metrics across all datasets and for both image and text modalities. For **Confidence Drop**, we measure the decrease in model confidence when low-attribution (i.e., unimportant) features are suppressed and only the high-attribution parts are retained; in this setting, a *smaller* drop indicates that the explanation has successfully captured most of the evidence the model relies on, and thus reflects better interpretability. ABM achieves the lowest Confidence Drop values on all datasets, particularly excelling on Conceptual Captions with a score of 0.7878 for images and a notably low 0.005019 for text. Similar trends are observed on ImageNet and Flickr8k, where

ABM attains 0.746 and 1.1169 for images, respectively, together with near-zero values for text. For **Confidence Increase**, we quantify how much the confidence *improves* when nonessential features are removed; here, larger values indicate that the removal of low-attribution content makes the model's decision more decisive and thus suggests a cleaner, more focused explanation. ABM again surpasses all baselines with the highest Confidence Increase across all datasets, achieving, for example, 57.1 for images and 60.4 for text on ImageNet.

In particular, compared with Grad-ECLIP, ABM substantially reduces the image-level Confidence Drop (e.g., from 2.3199 to 0.746 on ImageNet) while simultaneously increasing Confidence Increase (from 29.2 to 57.1), indicating that our adversarial bottleneck yields more faithful CLIP explanations than purely gradient-based saliency. Compared to M2IB, which is a strong competitor and widely acknowledged for its interpretability in multimodal tasks, ABM consistently shows better performance. On the Conceptual Captions dataset, ABM achieves a significantly lower Confidence Drop for both images (0.7878 vs. 1.1171) and text (0.005019 vs. 1.706), indicating more precise feature identification. Similarly, ABM excels on ImageNet with a Confidence Drop of 0.746 for images, compared to M2IB's 1.1615. In terms of Confidence Increase, ABM further demonstrates its superiority by achieving 57.1 for images and 60.4 for text on ImageNet, significantly outperforming M2IB's scores of 49.4 and 25.4. These results confirm that ABM offers more faithful and reliable attributions than M2IB across all tested datasets.

To further complement our evaluation, we include results under the **ROAD** (Rong et al., 2022) metric, which assesses the ability of an XAI method to discriminate between the most and least important pixels. A higher ROAD score indicates better attribution quality. As shown in Table 2, our ABM method again achieves the best overall score (1.8867), outperforming strong baselines such as Chefer et al. (1.8631), M2IB (1.7673), and FastIG (1.3211). Notably, methods like MFABA and Saliency Map exhibit negative ROAD scores, indicating poor discriminative ability in identifying key regions.

Table 2: Interpretability performance under the ROAD metric.

| Metric | ABM | M2IB | Chefer et al. | Grad-CAM | MFABA | RISE | Saliency Map | FastIG |
|--------|-----|------|---------------|----------|-------|------|--------------|--------|
| ROAD | **1.8867** | 1.7673 | 1.8631 | 1.5427 | -0.2851 | 0.0528 | -0.4216 | 1.3211 |

For the qualitative visual analysis, as shown in Figure 1, we compare the interpretability results of our ABM method with other baseline methods on the Conceptual Captions dataset. It is evident that our ABM method provides more precise and faithful explanations for both image and text modalities. In the first row, given the image of a traffic jam and the caption "cars drive along a main road," ABM accurately localizes key visual concepts such as cars and the road. Compared to other methods—e.g., FastIG and Saliency Map—that either overly diffuse attention across the image or miss the road structure, ABM clearly highlights the vehicle rows and central road divider, offering a more focused interpretation aligned with the text. In the second row, where the caption distinguishes between "anime" and "cartoons," ABM emphasizes the anime-style visual features in the lower-right corner while correctly associating them with the highlighted "anime" text tokens. In contrast, other methods either fail to disambiguate the visual-text alignment or activate irrelevant background regions, leading to noisy or misleading attribution. Overall, ABM consistently produces coherent and semantically aligned attributions for multimodal input, outperforming baselines in both spatial precision and cross-modal consistency. Additional qualitative examples are provided in **Appendix E**.

We also evaluated ABM and other baselines on AltCLIP (Chen et al., 2023)—a robust variant of CLIP—in **Appendix F.1**. The results demonstrate ABM's superior performance across the Conceptual Captions, ImageNet, and Flickr8k datasets. For computational efficiency analysis, we report the frames per second (FPS) of each method under the image modality in **Appendix F.2**. While achieving superior interpretability, our ABM demonstrates higher efficiency compared to the primary competing baseline, M2IB. Our ABM achieves the best overall results under this metric.

### 4.4 ABLATION STUDIES

We explore the impact of target layer $l$ and iteration count $T$ on ABM's image interpretability. As shown in Figure 2, the target layer substantially influences performance. When the target layer is set to 9, ABM shows a clear advantage over other layers across all datasets. Furthermore, $T$ also affects interpretability, with the best performance consistently at $T=10$. Specifically, for image explanations,

$T$=10 balances computational cost and attribution quality, yielding the strongest results. For text interpretability, results in **Appendix F.3** show ABM is not sensitive to these two hyperparameters, indicating stable and robust text explanations across settings. To provide more comprehensive ablations, we also include experiments in **Appendix F.4** evaluating early iterations ($T$=1, 2, 3, 4) for both image and text, revealing modality-specific differences in iteration sensitivity.

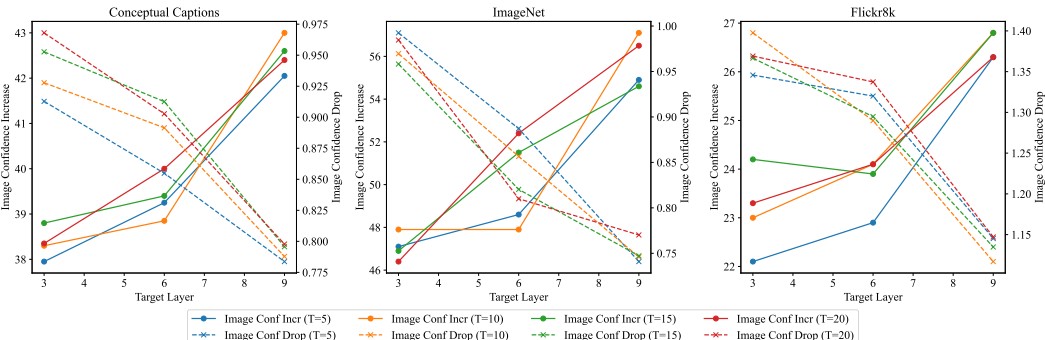

Figure 2: Impact of target layer and iteration count $T$ on ABM image interpretability across Conceptual Captions, ImageNet, and Flickr8k. Shown are Confidence Drop and Confidence Increase for target layers 3–9 and $T \in \{5, 10, 15, 20\}$. A higher Confidence Increase and a lower Confidence Drop indicate better interpretability. ABM attains its best performance at target layer 9 with $T = 10$.

## 5 CONCLUSION AND FUTURE WORK

In this paper, we tackle the interpretability challenges of large-scale vision-language models (VLMs), with a focus on CLIP. Existing methods like M2IB suffer from excessive sampling and hyperparameter sensitivity, while adversarial attribution techniques introduce redundancy. To address these issues, we propose the Adversarial Bottleneck Method (ABM)—an optimization of the Information Bottleneck Principle via adversarial perturbations that removes the need for tuning a sensitive trade-off hyperparameter such as $\beta$ and instead uses a robustness-insensitive iteration budget $T$ that acts as integration precision, thereby improving interpretability. ABM directly operates on single sample pairs, isolating critical features by maximizing cross-modal mutual information while minimizing input dependence. Theoretically, ABM compresses irrelevant information, and empirically, it outperforms baselines in both interpretability and efficiency. This work establishes ABM as a robust and reliable interpretability approach for VLM applications, opening pathways for its application to other multimodal models.

## ETHICS STATEMENT

We have read and will adhere to the ICLR Code of Ethics. This work uses only public data, involves no human subjects or personally identifiable information, and therefore does not require IRB review. Results are reported for research purposes only; we release anonymized code/configurations to support verification, and will disclose any funding sources and potential conflicts of interest upon acceptance.

## REPRODUCIBILITY STATEMENT

To support reproducibility, we release an anonymized repository with all experiment details including training/evaluation scripts, default hyperparameters, configuration files, and software/hardware environment.

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

## LLM USAGE DISCLOSURE

We used large language models (OpenAI GPT-4o and GTP-5) as auxiliary tools for grammar checking and language polishing of the manuscript. These models were not involved in research ideation, experimental design, implementation, or analysis. The authors take full responsibility for all content.

## A   PROOF OF THEOREM 3.1

*Proof.* We begin by defining $\tilde{z}^0$ as:

$$\tilde{z}^0 = \frac{z^0}{z} \cdot z + \left(1 - \frac{z^0}{z}\right) \cdot \varepsilon. \tag{13}$$

Since $z^0 = z$, it follows that

$$\tilde{z}^0 = z. \tag{14}$$

Moreover, $\tilde{z}^0 = z$ and $z = f(x_m)$ is a deterministic function of $x_m$, so $H(z \mid x_m) = 0$, and

$$I(\tilde{z}^0, x_m) = I(z, x_m) = H(z) - H(z \mid x_m) = H(z) = \max_{\tilde{z}} I(\tilde{z}, x_m), \tag{15}$$

and therefore

$$I(\tilde{z}^{t+1}, x_m) \leq I(\tilde{z}^0, x_m). \tag{16}$$

Next, define $\tilde{z}^{t+1}$ explicitly as a function of $z^t$:

$$\tilde{z}^{t+1} = h(z^t) = \frac{z^t}{z} \cdot z + \left(1 - \frac{z^t}{z}\right) \cdot \varepsilon. \tag{17}$$

From this definition, we have the exact identity

$$I(\tilde{z}^{t+1}, e_{m'}) = I(h(z^t), e_{m'}). \tag{18}$$

Under the standard first-order Taylor approximation (valid when the step size $\frac{z}{T}$ is sufficiently small), we then expand $I(h(z^t), e_{m'})$ around $z^t$:

$$I(\tilde{z}^{t+1}, e_{m'}) \approx I\big(h(z^t), e_{m'}\big) + (z^{t+1} - z^t)^\top \frac{\partial I(h(z^t), e_{m'})}{\partial z^t}, \tag{19}$$

where higher-order terms are omitted.

The update for $z^{t+1}$ is

$$z^{t+1} - z^t = C_z\left(\frac{z}{T} \cdot \text{sign}\left(\frac{\partial I(\tilde{z}^t, e_{m'})}{\partial z^t}\right)\right), \tag{20}$$

with element-wise operations and $T > 0$. Since $\frac{z}{T}$ is a positive scaling that does not change signs and $C_z(\cdot)$ is sign-preserving (it only clips magnitudes toward 0), we obtain

$$\text{sign}\left(C_z\left(\frac{z}{T} \cdot \text{sign}\left(\frac{\partial I(\tilde{z}^t, e_{m'})}{\partial z^t}\right)\right)\right) = \text{sign}\left(\frac{\partial I(\tilde{z}^t, e_{m'})}{\partial z^t}\right). \tag{21}$$

Hence,

$$(z^{t+1} - z^t)^\top \frac{\partial I(h(z^t), e_{m'})}{\partial z^t} > 0, \tag{22}$$

which implies

$$I(\tilde{z}^{t+1}, e_{m'}) > I(\tilde{z}^t, e_{m'}). \tag{23}$$

This completes the proof.  □

## B DERIVATION OF EQUATION 12 AND DETAILED PROOF

$$KL(p(\boldsymbol{x})\|q(\boldsymbol{x})) = \frac{1}{2}\left[(\boldsymbol{\mu}_p - \boldsymbol{\mu}_q)^\top \boldsymbol{\Sigma}_q^{-1}(\boldsymbol{\mu}_p - \boldsymbol{\mu}_q) - \log\det\left(\boldsymbol{\Sigma}_q^{-1}\boldsymbol{\Sigma}_p\right) + \mathrm{Tr}\left(\boldsymbol{\Sigma}_q^{-1}\boldsymbol{\Sigma}_p\right) - n\right] \tag{24}$$

$$A(z_i) = KL(P(\tilde{z}_i^T \mid x)\|\mathcal{N}(0,1))$$
$$= \frac{1}{2}\left(-1 - \log^{\left(1-\frac{z_i^T}{z}\right)^2} + \left(z_i^T\right)^2 + \left(1 - \frac{z_i^T}{z}\right)^2\right) \tag{25}$$

Equation 12 can be derived from Equation 24.

## C PSEUDOCODE OF THE ADVERSARIAL BOTTLENECK METHOD (ABM)

---
**Algorithm 1** ABM Explanation Algorithm
---
**Require:** Input model; intermediate-layer feature $z^0$
1: **Init:** $z^0 = z$
2: **for** $t = 0, \ldots, T-1$ **do**
3:      $\tilde{z}^t = z^t + \left(1 - \frac{z^t}{z}\right) \cdot \varepsilon, \qquad \varepsilon \sim \mathcal{N}(0, I)$
4:      Use $\tilde{z}^t$ (or $z^{t+1}$) to replace $z$ for forward propagation
5:      $z^{t+1} = C_z\left(z^t + \frac{z}{T} \cdot \mathrm{sign}\left(\frac{\partial I(\tilde{z}^t, e_{m'})}{\partial z^t}\right)\right)$
6: **end for**
7: **for** $i = 0, \ldots, |z|$ **do**
8:      $A(z_i) = \mathrm{KL}\left(P(\tilde{z}_i^T \mid x) \,\middle\|\, \mathcal{N}(0,1)\right)$
9:      $= \frac{1}{2}\left(-1 - \log\left(1 - \frac{z_i^T}{z}\right)^2 + (z_i^T)^2 + \left(1 - \frac{z_i^T}{z}\right)^2\right)$
10: **end for**
11: **return** $A(z)$
---

## D BOOLEAN EVALUATION FOR TEXT MODALITY

When evaluating the model's text data, we followed the evaluation method used in M2IB. Specifically, in the interpretability evaluation of the text modality, a Boolean value is used as the evaluation criterion. This design ensures that the text evaluation results are based on binary judgments, i.e., whether the confidence change after perturbation is greater than zero, rather than involving continuous value changes. Therefore, changes in the number of iterations $T$ do not affect this Boolean-based evaluation. This characteristic directly leads to the invariance of the text interpretability results in the table across different $T$ settings. The evaluation code is as follows:

```python
class DropInConfidenceText(CamMultImageConfidenceChange):
    def __init__(self):
        super(DropInConfidenceText, self).__init__()

    def __call__(self, *args, **kwargs):
        scores = super(DropInConfidenceText, self).__call__(*args,
                                                            **kwargs)
        scores = -scores
        return np.maximum(scores, 0)

class IncreaseInConfidenceText(CamMultImageConfidenceChange):
    def __init__(self):
        super(IncreaseInConfidenceText, self).__init__()

    def __call__(self, *args, **kwargs):
```

```
756        scores = super(IncreaseInConfidenceText, self).__call__(*args,
757                                                    **kwargs)
758        return np.float32(scores > 0)
```

# E  ADDITIONAL QUALITATIVE VISUALIZATION EXPLANATIONS

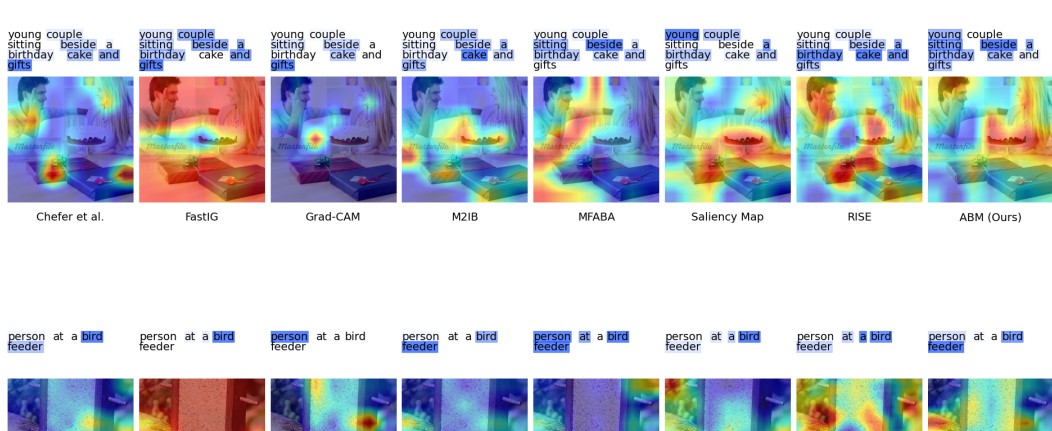

Figure 3: Additional qualitative visualization explanations on Conceptual Captions dataset.

# F  ADDITIONAL EXPERIMENTS

## F.1  EXPERIMENT RESULTS ON ALTCLIP MODEL

We conducted additional experiments on AltCLIP, demonstrating that our ABM consistently maintains the best interpretability performance. The Table 3 indicates that our method outperforms other approaches across different model variants.

## F.2  COMPUTATIONAL EFFICIENCY

We conducted experiments measuring the inference speed in terms of frames per second (FPS) on the CLIP model using the Conceptual Captions dataset, all tested on an Nvidia A100 GPU.

Table 4 demonstrates that ABM achieves a significant improvement in interpretability performance while maintaining competitive inference speed compared to similar methods. Specifically, ABM outperforms M2IB—a method also designed for CLIP interpretability—achieving a higher FPS (1.67 vs. 1.02), highlighting its efficiency. Although methods like FastIG and SaliencyMap exhibit higher FPS, these methods do not provide the same level of interpretability performance as ABM.

## F.3  ADDITIONAL ABLATION STUDY ON TEXT INTERPRETABILITY OF ABM

Unlike the image modality, ABM's performance for text interpretability is not sensitive to hyperparameter adjustments. As shown in Table 5, the number of iterations $T$ does not affect the interpretability of ABM for text at all. Furthermore, while the choice of target layer has some impact, it is only minimal. The performance remains consistent across different layers and number of iterations configurations, indicating that ABM's text explanations are stable and robust regardless of the hyperparameter changes.

Table 3: Experiment results on AltCLIP model

| Dataset | Method | Image | | Text | |
|---|---|---|---|---|---|
| | | **Conf Drop (↓)** | **Conf Incr (↑)** | **Conf Drop (↓)** | **Conf Incr (↑)** |
| Conceptual Captions | M2IB | 1.3324 | 32.0 | 4.8767 | 22.25 |
| | RISE | 1.3115 | 27.6 | 1.6627 | 32.95 |
| | Grad-CAM | 6.6955 | 9.6 | 8.0304 | 8.9 |
| | Chefer et al. | 5.3088 | 13.2 | 6.2125 | 11.30 |
| | SaliencyMap | 1.2415 | 32.5 | 2.6278 | 24.35 |
| | FastIG | 1.8274 | 26.8 | 2.6237 | 30.65 |
| | ABM(Ours) | **0.9183** | **36.8** | **0.0020** | **47.65** |
| ImageNet | M2IB | 1.6074 | **45.5** | 5.7696 | 4.4 |
| | RISE | 2.0764 | 35.2 | 4.6092 | 4.2 |
| | Grad-CAM | 8.1205 | 5.8 | 7.0187 | 3.7 |
| | Chefer et al. | 7.0399 | 6.8 | 7.2695 | 3.4 |
| | SaliencyMap | 1.7094 | 40.0 | 3.5131 | 11.1 |
| | FastIG | 1.8590 | 39.0 | 4.1739 | 11.8 |
| | ABM(Ours) | **1.2385** | 45.0 | **0.0076** | 21.9 |
| Flickr8k | M2IB | 1.2767 | 28.4 | 3.8204 | 13.0 |
| | RISE | 1.5285 | 19.0 | 2.2361 | 20.5 |
| | Grad-CAM | 7.5028 | 6.2 | 9.0013 | 2.5 |
| | Chefer et al. | 5.5111 | 11.1 | 9.3455 | 1.3 |
| | SaliencyMap | 1.4095 | 24.7 | 2.6532 | 19.8 |
| | FastIG | 2.5895 | 12.6 | 2.6520 | 25.1 |
| | ABM(Ours) | **0.8920** | **33.2** | **0.0042** | **40.7** |

Table 4: Computational efficiency experimental results

| Method | M2IB | ABM | Chefer et al. | FastIG | MFABA | RISE | Grad-CAM | SaliencyMap |
|---|---|---|---|---|---|---|---|---|
| FPS | 1.02 | 1.67 | 5.83 | 6.05 | 0.96 | 0.02 | 4.79 | 6.36 |

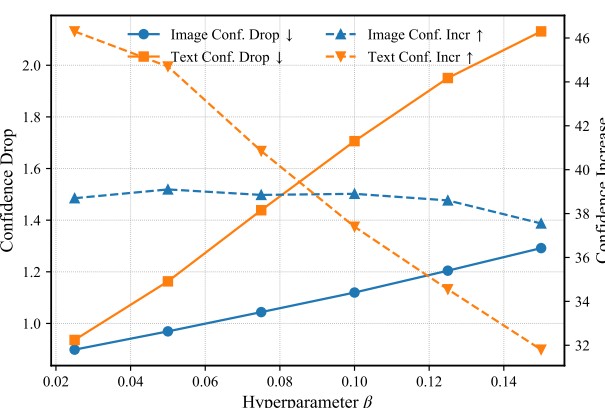

Figure 4: Performance sensitivity of M2IB with respect to the hyperparameter $\beta$ on the Conceptual Captions dataset using the CLIP model. The plot illustrates the trends of four evaluation metrics—Image/Text Confidence Drop (↓) and Image/Text Confidence Increase (↑)—as $\beta$ varies. The results show that M2IB exhibits significant performance fluctuations across modalities, indicating that the choice of $\beta$ has a substantial impact on model behavior.

## F.4 ADDITIONAL ABLATION STUDY ON HYPERPARAMETER $T$ BEFORE 5

We have conducted additional ablation experiments to provide a more comprehensive analysis of the model's behavior across different numbers of iterations ($T$) for both image and text modalities. These experiments include evaluations for $T = 1, 2, 3, 4$ across various datasets (Conceptual Captions, Flickr8k, ImageNet) and target layers (3, 6, 9).

As shown in the Table 6, for text modality, the interpretability metrics Confidence Drop and Confidence Increase remain consistent across all $T$ settings. This invariance is due to the evaluation design, which employs binary judgments (as inspired by the M2IB method). In contrast, for the image modality, the metrics exhibit noticeable variations with changes in $T$. These differences stem

Table 5: The impact of target layer and number of iterations $T$ on ABM's text interpretability across three datasets (Conceptual Captions, ImageNet, Flickr8k). The table shows the performance based on **Confidence Drop** ↓ and **Confidence Increase** ↑ metrics. A lower Confidence Drop and a higher Confidence Increase indicate better interpretability. The results demonstrate that for text tasks, both the number of iterations and target layer choices have negligible influence on ABM's interpretability.

| Target Layer | Iterations Number $T$ | Conceptual Captions | | ImageNet | | Flickr8k | |
|---|---|---|---|---|---|---|---|
| | | Text Conf Drop ↓ | Text Conf Incr ↑ | Text Conf Drop ↓ | Text Conf Incr ↑ | Text Conf Drop ↓ | Text Conf Incr ↑ |
| 3 | 5 | 0.00098 | 44.45 | 0.00072 | 62.9 | 0.00090 | 58.2 |
| 6 | 5 | 0.00252 | 44.55 | 0.00182 | 60.3 | 0.00203 | 59.2 |
| 9 | 5 | 0.00502 | 44.5 | 0.00491 | 60.4 | 0.00390 | 59.3 |
| 3 | 10 | 0.00098 | 44.45 | 0.00072 | 62.9 | 0.00090 | 58.2 |
| 6 | 10 | 0.00252 | 44.55 | 0.00182 | 60.3 | 0.00203 | 59.2 |
| 9 | 10 | 0.00502 | 44.5 | 0.00491 | 60.4 | 0.00390 | 59.3 |
| 3 | 15 | 0.00098 | 44.45 | 0.00072 | 62.9 | 0.00090 | 58.2 |
| 6 | 15 | 0.00252 | 44.55 | 0.00182 | 60.3 | 0.00203 | 59.2 |
| 9 | 15 | 0.00502 | 44.5 | 0.00491 | 60.4 | 0.00390 | 59.3 |
| 3 | 20 | 0.00098 | 44.45 | 0.00072 | 62.9 | 0.00090 | 58.2 |
| 6 | 20 | 0.00252 | 44.55 | 0.00182 | 60.3 | 0.00203 | 59.2 |
| 9 | 20 | 0.00502 | 44.5 | 0.00491 | 60.4 | 0.00390 | 59.3 |

Table 6: Additional ablation study results on hyperparameter $T$ before 5

| Dataset | Target layer | steps $T$ | Image | | Text | |
|---|---|---|---|---|---|---|
| | | | Image Conf Drop | Image Conf Incr | Text Conf Drop | Text Conf Incr |
| Conceptual Captions | 3 | 1 | 0.864 | 39.6 | 0.000983 | 44.5 |
| | 3 | 2 | 0.922 | 37.55 | 0.000983 | 44.5 |
| | 3 | 3 | 0.913 | 38.25 | 0.000983 | 44.5 |
| | 3 | 4 | 0.895 | 37.85 | 0.000983 | 44.5 |
| | 6 | 1 | 0.784 | 41.6 | 0.002515 | 44.55 |
| | 6 | 2 | 0.880 | 37.9 | 0.002515 | 44.55 |
| | 6 | 3 | 0.853 | 39.85 | 0.002515 | 44.55 |
| | 6 | 4 | 0.879 | 38.75 | 0.002515 | 44.55 |
| | 9 | 1 | 0.770 | 43.7 | 0.005019 | 44.4 |
| | 9 | 2 | 0.785 | 41.5 | 0.005019 | 44.4 |
| | 9 | 3 | 0.785 | 43.45 | 0.005019 | 44.4 |
| | 9 | 4 | 0.770 | 41.35 | 0.005019 | 44.4 |
| Flickr8k | 3 | 1 | 1.398 | 24 | 0.000947 | 57.8 |
| | 3 | 2 | 1.442 | 21 | 0.000947 | 57.8 |
| | 3 | 3 | 1.395 | 21.8 | 0.000947 | 57.8 |
| | 3 | 4 | 1.381 | 22.5 | 0.000947 | 57.8 |
| | 6 | 1 | 1.243 | 26.6 | 0.002199 | 60.4 |
| | 6 | 2 | 1.381 | 20.9 | 0.002199 | 60.4 |
| | 6 | 3 | 1.303 | 25.6 | 0.002199 | 60.4 |
| | 6 | 4 | 1.299 | 23.4 | 0.002199 | 60.4 |
| | 9 | 1 | 1.260 | 25.3 | 0.004142 | 60.8 |
| | 9 | 2 | 1.200 | 25.9 | 0.004142 | 60.8 |
| | 9 | 3 | 1.168 | 26.3 | 0.004142 | 60.8 |
| | 9 | 4 | 1.153 | 26.3 | 0.004142 | 60.8 |
| ImageNet | 3 | 1 | 0.835 | 47.1 | 0.000720 | 62.9 |
| | 3 | 2 | 1.035 | 42.4 | 0.000720 | 62.9 |
| | 3 | 3 | 1.030 | 45.1 | 0.000720 | 62.9 |
| | 3 | 4 | 0.999 | 44 | 0.000720 | 62.9 |
| | 6 | 1 | 0.878 | 48.8 | 0.001824 | 60.3 |
| | 6 | 2 | 0.931 | 46.5 | 0.001824 | 60.3 |
| | 6 | 3 | 0.957 | 47.1 | 0.001824 | 60.3 |
| | 6 | 4 | 0.848 | 50.2 | 0.001824 | 60.3 |
| | 9 | 1 | 0.873 | 49.7 | 0.004906 | 60.4 |
| | 9 | 2 | 0.752 | 53.8 | 0.004906 | 60.4 |
| | 9 | 3 | 0.754 | 55.5 | 0.004906 | 60.4 |
| | 9 | 4 | 0.721 | 56.4 | 0.004906 | 60.4 |

from the continuous evaluation metrics used for images, which are more sensitive to iterative updates and better capture the refinement of feature importance with increasing $T$.

## F.5   Sensitivity to the Hyperparameter $\beta$

To investigate the sensitivity of M2IB to the hyperparameter $\beta$, we conducted a controlled experiment on the Conceptual Captions (CC) dataset using the CLIP model. As shown in Figure 4, M2IB exhibits considerable performance fluctuation with respect to different $\beta$ values. Although the default setting in the original paper is $\beta = 0.1$, our results indicate that this may not be the optimal choice. Specifically, we observe that the confidence drop and confidence increase metrics for both image and text modalities vary significantly as $\beta$ changes. For instance, smaller values of $\beta$ tend to achieve

lower confidence drop scores for image features, whereas intermediate values (around $\beta = 0.08$) offer more balanced performance across modalities. The inconsistency and sharp variations across all four metrics suggest that M2IB is highly sensitive to the selection of $\beta$, which raises concerns about the robustness and generalizability of the method under different hyperparameter settings.

