# OpenReview forum: "Adversarial Bottleneck Method for Vision-Language Large Model Explainability"
_ICLR.cc/2026/Conference — Submitted to ICLR 2026_

### Official Review · Reviewer_P2P4 · 2025-10-24

**Soundness:** 2
**Presentation:** 2
**Contribution:** 2
**Rating:** 4
**Confidence:** 4

**Summary:**

This paper introduces ABM, a VLM interpretation method that works by fusing adversarial attribution and information bottleneck. ABM improves over M2IB by performing adversarial updates on intermediate-layer bottleneck variables with a sign-based step scaled by the original magnitude and clipped by a constraint, which overcomes the limitation of M2IB including sensitivity to hyper-parameters and difficulty in gradient-based optimization.

**Strengths:**

1. The idea to integrate adversarial attribution and information bottleneck for VLM explanation is interesting.
2. The authors provide theoretical insights into the proposed method.
3. The evaluation is comprehensive.

**Weaknesses:**

1. The impact of this paper is small to me. It improves over M2IB, a very specific VLM interpretation technique. And the improvement mainly focuses on the optimization of M2IB.  M2IB is not the SOTA method for CLIP interpretation, with later work such as Grad-ECLIP [1] showing much better performance.
2. The readability of the math part is poor.  Many symbols are reused to represent different meanings (e.g., $T$ represents both text and the total steps, $g$ appears in both eq.3 and eq.16 with different meanings). And many clarifications are posited far away from the equation (e.g., the explanation in line 315 should be put earlier). This makes the paper very difficult to read.
3. ABM is built on M2IB, but the paper does not introduce the specific algorithm or derivation of M2IB, only a high-level introduction of M2IB is given on Sec.3.2.1. This makes the paper less self-contained. For example, when introducing eq.11, the KL term seems very strange as it has not been mentioned before. Also, I would recommend the authors to include an algorithm box to better clarify ABM.
4. I do not totally understand the proof of theorem3.1. There are some typos. For example, in eq.16, it states, $\tilde{z}^{t+1}=g(z^t)$, then in 17, why not $I(\tilde{z}^{t+1}, e_{m’})= I(g(z^t), e_{m’})$? Moreover, how can you entirely ignore the higher-order terms when proving the monotonicity?
5. In line 372, the definition of confidence drop is confusing. Shouldn’t a larger drop confidence drop when removing the important features denote better interpretability? The correct definition should be the drop in performance if only the high-attribution parts are kept.

[1] Zhao, Chenyang, et al. "Gradient-based visual explanation for transformer-based clip." ICML 2024.

**Questions:**

1. Could you compare your method with Grad-ECLIP?

---

> ### Author Response · Authors · 2025-11-22
> **Reply to Weaknesses 1, 2, 3 and Question 1**
>
> **Reply to W1 & Q1:** We appreciate the reviewer’s comment and the pointer to more recent CLIP interpretation methods. In response, we have added Grad-ECLIP [1] as an additional strong baseline in the revised manuscript. The corresponding quantitative results are reported in the table below and have been integrated into Table 1 of the main paper.
>
> |              | Conceptual Captions | Conceptual Captions | Conceptual Captions | Conceptual Captions | ImageNet | ImageNet | ImageNet | ImageNet | Flickr8k | Flickr8k | Flickr8k | Flickr8k |
> |--------------|---------------------|---------------------|---------------------|---------------------|----------|----------|----------|----------|----------|----------|----------|----------|
> |              | Image               | Image               | Text                | Text                | Image    | Image    | Text     | Text     | Image    | Image    | Text     | Text     |
> |              | Conf Drop           | Conf Incr           | Conf Drop           | Conf Incr           | Conf Drop| Conf Incr| Conf Drop| Conf Incr| Conf Drop| Conf Incr| Conf Drop| Conf Incr|
> | Grad-ECLIP   | 2.3956              | 26.5                | 1.2894              | 43.2                | 2.3199   | 29.2     | 2.0104   | 28.3     | 7.7444   | 2.6      | 2.1114   | 37.5     |
> | ABM (Ours)   | 0.7878              | 43.0                | 0.005019            | 44.5                | 0.746    | 57.1     | 0.0049   | 60.4     | 1.1169   | 26.8     | 0.0039   | 59.3     |
>
> As the numbers show, our ABM method consistently achieves **lower Confidence Drop** and **higher Confidence Increase** than Grad-ECLIP across all three datasets and for both image and text modalities. For example, on ImageNet, ABM reduces the image-level Confidence Drop from 2.3199 (Grad-ECLIP) to 0.746 and boosts Confidence Increase from 29.2 to 57.1; on Flickr8k, ABM lowers the image-level Confidence Drop from 7.7444 to 1.1169 and substantially improves the text-level Confidence Increase from 37.5 to 59.3. These results indicate that, beyond improving over M2IB, the proposed adversarial bottleneck framework also outperforms a recent gradient-based CLIP interpretation method, suggesting that its impact is not limited to optimizing a single prior technique but provides a more faithful and general mechanism for CLIP attribution.
>
> **Reference**
>
> [1] Zhao, Chenyang, et al. "Gradient-based visual explanation for transformer-based clip." ICML 2024.
>
> ---
>
> **Reply to W2:**
> We thank the reviewer for pointing out these clarity issues in the mathematical notation. We have corrected all of them in the revised manuscript.
> Specifically, in Section 3.1 we now use $\mathcal{T}$ to replace the previous symbol $T$ in order to clearly distinguish the text token from the total number of update steps. In addition, the construction function $g$ in Equation (16) has been renamed to $h$ to avoid overloading notation.
>
> We have also moved the clarification previously located at line 315—namely,
> “$T$ represents the number of update steps and $z^0 = z$. The vector $z$ denotes the latent representation, and the ratio $\frac{z^t}{z}$ is taken element-wise.”, so that it now appears directly after Theorem 3.1, where it belongs and improves readability.
>
> ---
>
> **Reply to W3:**
> Regarding the KL term in Eq. (11) (Eq. (12) in the revised manuscript): ABM adopts a Gaussian bottleneck prior $\mathcal{N}(0, I)$, and this term measures, for each latent dimension, how far the final bottleneck distribution $P(\tilde{z}_i^T \mid x)$ deviates from this noise prior. A larger deviation indicates that more useful information is being preserved along that dimension, and thus that dimension is more important. We will move the discussion of this prior and its role in the bottleneck to appear before this equation, so that the KL term does not seem to “appear out of nowhere.” In addition, we have added a pseudocode-style algorithm block for ABM in the appendix to further clarify the method.

---

> ### Author Response · Authors · 2025-11-22
> **Reply to Weaknesses 4, 5**
>
> **Reply to W4:**
> Thank you for pointing this out. We agree that the transition from Eq. (16) (Eq. 17 in the revised version) to Eq. (17) (Eq. 19 in the revised version) was not written explicitly enough, which caused confusion. In the revision, we now state this step fully and directly. Since Eq. (18) defines  $\tilde{z}^{t+1} = h(z^t),$ it immediately follows that $I(\tilde{z}^{t+1}, e_{m'}) = I(h(z^t), e_{m'}).$
>
> We have inserted this exact identity before the Taylor expansion, ensuring notational consistency and a clear logical flow. No mathematical changes are made—the revision only adds the missing intermediate step to eliminate ambiguity.
>
> Regarding the omission of higher-order terms: when the step size is sufficiently small, higher-order terms can be ignored and the gradient provides the correct ascent direction. We will explicitly add the condition of “sufficiently small step size” in the revision. This is the standard and foundational assumption behind neural-network optimization and is necessary for gradient-based training itself (without this assumption, gradient descent for neural networks would not hold).
>
> ---
>
> **Reply to W5:** Regarding the definition of Confidence Drop (line 372), we agree that our original wording was confusing and could be interpreted as “removing important features”. In our implementation, however, Confidence Drop is computed in a way that is closer to what the reviewer suggests: we keep the high-attribution (important) parts and suppress the low-attribution (unimportant) features, and then measure how much the model’s confidence decreases under this restricted input. In this setting, a smaller Confidence Drop means that the explanation has captured most of the evidence the model relies on (i.e., the model can still maintain its confidence when only the highlighted features are present), so lower values indicate better interpretability.
>
> To avoid this confusion, we have revised the text around line 372 to explicitly state that Confidence Drop is defined as the confidence decrease “when low-attribution features are suppressed and only the high-attribution parts are retained”, instead of “when important features are removed”. We have also clarified that Confidence Increase is defined as the confidence gain when nonessential features are removed, where larger values indicate that the explanation successfully filters out distracting content. These clarifications better align the description in the paper with the actual implementation.
>
> ---
>
> Thanks Reviewer P2P4 for careful reading and detailed comments on our work. In the revision, we have (i) broadened the empirical scope by adding Grad-ECLIP as a strong, recent CLIP interpretation baseline and shown that ABM consistently outperforms it across datasets and modalities, (ii) revised the notation and reordered explanations to improve the readability of the mathematical sections, and (iii) expanded the M2IB background and added a pseudocode-style algorithm block so that the paper is more self-contained and easier to follow. We have also clarified the proof steps in Theorem 3.1 and the precise definitions of Confidence Drop/Increase to match the implementation. We hope these changes address your concerns about impact, clarity, and presentation, and we respectfully ask you to reconsider your evaluation in light of our replies and the revised manuscript. If any part of our response remains unclear, we would of course be happy to clarify further.

---

### Official Review · Reviewer_WHDb · 2025-10-29

**Soundness:** 1
**Presentation:** 1
**Contribution:** 1
**Rating:** 2
**Confidence:** 4

**Summary:**

The authors study the interpretability of vision language models (VLMs) and attempt to improve the existing M2IB method (reviewed in equation 2) by the framework called "Adversarial Attribution Theory". In Sec. 3.2.1. the authors argue that the M2IB method may give untrustworthy results because of the fixed choice of hyperparameter $\beta$ and the hyperparameter choice of the gradient descent method (Lines 216-220). Then, it seems the authors review some calculus in section 3 to propose a new optimization approach, by introducing the extra $C_z$ clipping operator. They present numerical results in section 4.

**Strengths:**

-The authors' motivation to improve the interpretability of CLIP is sensible.

**Weaknesses:**

The paper has several flaws in its theoretical claims, technical soundness, and presentation.

-**Incorrect theoretical result in Theorem 3.1**: Theorem 3.1, the paper's only theorem, tries to prove Equations 7 and 8. However, the proof of Equation 7 is incorrect and suffers from a basic mistake. In the Appendix sec A, the authors mention the following equation in their proof (Lines 663–665):

> $$I(\tilde{z}^0, x_m) = H(x_m) − H(z | x_m) = H(x_m)$$

The above equation is wrong and the correct equation is $I(Z;X)= H(X) - H(X|Z)$, where the conditional entropy is mistakenly stated to be $H(Z|X)$ by the authors. This error invalidates the authors’ claim, because they use the wrong equation to claim $H(z | x_m) = 0$. This is while, in the correct form, $H(x_m|z) \neq 0 $ unless $x_m$ is a function of $z$, being exactly the opposite of the authors' statement in front of Equation 13 and disproving their claim in Equation 7.

-**Trivial theoretical claim in Equation 8 of Theorem 3.1**: In addition to the wrong statement in Equation 7, the authors state a trivial result in Equation 8 (second part of Theorem 3.1). If I understood the claim correctly (the presentation is unclear), Equation 8 says that, assuming a zero error for the first-order Taylor series expansion (as supposed in Line 671 and not mentioned in Theorem 3.1), applying the gradient ascent with the clipped gradient in Equation (9) will only increase the function value.

However, I believe this is a trivial statement under the assumption of zero error for the first-order Taylor series expansion. Given the first-order Taylor expansion, the function value will trivially increase locally along any direction with a positive inner product with the gradient direction, and the clipping operation is the projection of the gradient vector on the $\ell_\infty$-norm ball, which will obviously have a positive inner product with the gradient direction. I am wondering in what sense Equation (8) in Theorem 3.1 proves something non-trivial, as this part of the theorem is the only remaining theoretical claim in the draft, excluding the wrong Equation (7).

-**Clarity of the presentation**: I cannot understand the purpose of Equations 3 and 4. Equations 3 and 4 seem to state the fundamental theorem of calculus, that if one integrates the gradient of a function along a path from $z^0$ to $z^T$, the output is the function difference evaluated at $z^0$ and $z^T$. I cannot appreciate why the authors spend more than half a page on this basic fact. In what sense does this discussion support the known optimization iteration (projected gradient ascent using $\ell_\infty$-norm ball) in Equation 5?

-**Presentation issues**: There are several writing errors in the text. Some obvious cases are the misspecified references “(?)” in Lines 182–184 and the wrong in-line position of the $\log$ input in Equation 11.

Also, I could not see how the authors obtained Equation (10) from what they discussed before. $z^{(t)}$ is supposed to follow the update Equation (6) with the mutual information between the CLIP input and output variables. How does this equation lead to Equation (10)? What is the role of the mutual information in getting the update rule in (10) from Equation (6)? As said, it is difficult to follow the discussion after this equation due to the incoherence in writing and notations.

**Questions:**

1. Can the authors clarify the correctness of Equation (7) and the non-triviality of Equation (8) in Theorem 3.1, which appears to be the main technical claim of this work?

2. How do the authors derive the update rule (10) from Equations (6) and (9)?

3. The authors criticize the M2IB method of Wang et al. (2023) in Equations (1) and (2) for relying on the $\beta$ hyperparameter in Equation (2) and for using gradient ascent to solve (2) (as stated in Lines 210–213). Can the authors clearly explain how their algorithm differs from that of Wang et al. (2023)? Specifically, is the only modification the introduction of the clipping operator in Equations (6) and (9) in place of the vanilla gradient ascent update?

---

> ### Author Response · Authors · 2025-11-22
> **Reply to Weaknesses 1, 2 and Question 1**
>
> **Reply to W1:** We thank the reviewer for the careful examination of the proof of Theorem 3.1. You are correct that the mutual-information identity in Appendix A was written incorrectly; however, this was a typographical mistake in which we accidentally wrote $z$ as $x_m$. In the current manuscript (Equation (14)), we wrote:$I(\tilde{z}^0, x_m) = H(x_m) - H(z \mid x_m) = H(x_m) = \max I(\tilde{z}, x_m)$
>
> The correct identity should be:$I(Z; X) = H(Z) - H(Z \mid X) = H(X) - H(X \mid Z).$
>
> In our setting, $\tilde{z}_0 = z$ and $z = f(x_m)$ is a deterministic function of $x_m$. Therefore, $H(z \mid x_m) = 0.$
>
> Hence, the line above should be rewritten as:$I(\tilde{z}^0, x_m) = I(z, x_m) = H(z) - H(z \mid x_m) = H(z) = \max I(\tilde{z}, x_m),$
>
> and we never assumed $H(x_m \mid z) = 0$. The phrase “uniquely confirmed by (x_m)” in the appendix is likewise misleading, and we will remove it.
>
> This correction fixes a presentation error in the proof but does **not** affect the intended inequality in Equation (7): by construction, $\tilde{z}_t$ is obtained by progressively replacing coordinates of $z$ with independent Gaussian noise, and therefore $I(\tilde{z}^{t+1}, x_m) \le I(\tilde{z}^0, x_m)$ still holds. We have corrected the erroneous notation in the revised version.
>
> ---
>
> **Reply to W2 & Q1:** When assuming a first-order Taylor approximation and a sufficiently small step size, the inequality in Equation (8) is indeed conceptually just the standard local property of gradient ascent: moving along any direction that has a positive inner product with the gradient will increase the objective value (ignoring higher-order terms), and projecting this direction onto a bounded-norm set does not break this property.
>
> There are two points we would like to clarify:
>
> **1. Regarding the assumption of the Taylor approximation.**
> Our intention was not to assume that the first-order Taylor expansion has “zero error,” but rather to adopt the standard local analysis widely used in optimization and adversarial attribution: when the step size is sufficiently small, higher-order terms can be ignored, and the gradient indicates the ascent direction. We will explicitly add the condition that the step size should be sufficiently small. This is the most basic assumption behind neural network optimization and a necessary condition for gradient descent training (without this assumption, gradient descent for neural networks would not hold).
>
> **2. Regarding the role and non-triviality of Equation (8).**
> We agree that Equation (8) is not mathematically “deep” by itself; its purpose is to formalize an important point for our constrained adversarial update:
>
> $$
> z^{t+1} = C_z\left(z^t + \frac{z}{T} \cdot \operatorname{sign}\left(\frac{\partial I(\tilde{z}^t, e_{m'})}{\partial z^t}\right)\right)
> $$
>
> namely, that it is still an ascent step for
> $ I(\tilde{z}^t, e_{m'}) $ under a first-order view, despite using element-wise sign steps and the clipping operator $ C_z$. Therefore, the goal of Theorem 3.1 is not to claim a strong new theoretical result about gradient methods, but to show that this particular **adversarial bottleneck update** simultaneously satisfies two properties under the first-order approximation:
>
> (i) it pushes each coordinate toward its noise endpoint, thereby compressing information about the sample;
>
> (ii) it remains aligned with the IB objective by monotonically increasing $I(\tilde{z}^t, e_{m'}) $.
>
> We emphasize in the revision that our main contributions lie in the overall modeling of ABM and its empirical behavior.

---

> ### Author Response · Authors · 2025-11-22
> **Reply to Weaknesses 3, 4 and Question 2**
>
> **Reply to W3:** At the level of basic calculus, Equations (3) and (4) indeed restate a standard fact: integrating the gradient of a function along a path gives the difference $L(z^T) - L(z^0)$. However, our intention is not to present this as a new analysis of gradient ascent, but to use it to formalize two specific aspects of adversarial attribution:
>
> **(1) Decomposing the loss difference into per-feature attribution scores.**
> Equations (3)–(4) show that the change in loss along the update path can be written as
> $$
> \sum_{i=1}^D \int \Delta z_i^t \odot g(z_i^t), dt = L(z^T) - L(z^0).
> $$
> We interpret this as assigning to each latent feature $i$ a contribution term
> $$
> \int \Delta z_i^t \odot g(z_i^t), dt,
> $$
> which is precisely the quantity we use to define the “attribution score” of dimension $i$ in adversarial attribution. The purpose of Equations (3)–(4) is to make explicit that our attribution is *by construction* a per-dimension decomposition of the overall loss difference.
>
> **(2) Connecting this decomposition to the Sensitivity and Completeness axioms.**
> Under this formulation, if a feature’s contribution term is always zero along all admissible paths, then changing that feature cannot affect the loss—corresponding to the Sensitivity axiom in attribution methods. Meanwhile, summing the contributions over all features recovers
> $$
> L(z^T) - L(z^0),
> $$
> which corresponds to the Completeness axiom. Therefore, Equations (3)–(4) show that our path-based adversarial attribution naturally satisfies Sensitivity and Completeness by design, rather than imposing these properties heuristically afterward.
>
> Equation (5) then specifies the particular path used in practice: a projected gradient-ascent adversarial step with an $\ell_\infty$ constraint. In other words, (3)–(4) define how any such path induces per-feature attributions and guarantees Sensitivity/Completeness, while (5) chooses a concrete path for our setting.
>
> The additional intermediate line in the derivation helps clarify this process. We will add a brief explanation of these two points in the revised manuscript to aid understanding.
>
> ---
>
> **Reply to W4 & Q2:** First, we thank the reviewer for carefully pointing out the writing errors in the manuscript. We have corrected these issues in the revised version.
>
> Regarding the derivation of Equation (10) from Equation (6), conceptually our construction actually consists of two distinct steps, and we will add explanations to clearly separate them:
>
> **Step 1: Introduce the gated latent variable $ z^t $ and update it using mutual information.**
> Equation (6) defines the update rule for an auxiliary *gating* variable $ z^t$. This variable is constrained by the operator $C_z$ so that it increases the mutual information:
>
> $$
> z^{t+1} = C_{z}\left(z^{t} + \frac{z}{T} \cdot \operatorname{sign}\left(\frac{\partial I(\tilde{z}^{t}, e_{m'})}{\partial z^{t}}\right)\right)
> $$
>
> Here, the mutual information $I(\tilde{z}^t, e'_{m})$ only appears through its gradient: it tells us in which direction each dimension of $z^t$ should move to increase cross-modal dependence. The operator $C_z$ and the factor $z/T$ ensure that each coordinate always stays between its original value $z$ and $0$.
>
> **Step 2: Use the updated $z^t$ to control how much information passes through the bottleneck.**
> (For clarity, let me introduce a gate variable $\gamma_t$.)
>
> $$
> \tilde{z}^t = \frac{z^t}{z} \cdot z + \left(1 - \frac{z^t}{z}\right)\cdot \varepsilon
> = z^t + \left(1 - \frac{z^t}{z}\right)\cdot \varepsilon, \qquad \varepsilon \sim \mathcal{N}(0, I).
> $$
>
> In other words, we first determine—based on the Information Bottleneck goal—how $\tilde{z}^t$ should interpolate between the “full-information representation” $z$ (no compression) and “independent Gaussian noise” $\varepsilon$ (maximum compression). This naturally corresponds to a gate variable $\gamma_t \in [0,1]$ such that:
>
> $$
> \tilde{z}^t = \gamma_t z + (1 - \gamma_t)\varepsilon.
> $$
>
> Therefore, Equation (10) is simply the mapping from the updated gate $z^t$ to the actual bottleneck representation $\tilde{z}^t$.
>
> **Overall:**
> The role of mutual information is to define the update direction of $z^t$ in Equation (6) through its gradient so that $I(\tilde{z}^t, e_{m'})$ increases.
>
> Equation (10), on the other hand, is an IB-motivated parametrization: it expresses $\tilde{z}^t$ as a mixture of $z$ and noise, with the mixture controlled by $z^t$ (or equivalently the gate $\gamma_t$). Equation (10) is not intended to be “derived” from a closed-form expression of mutual information. Without Equation (10), $z^t$ would not influence the model at all, and thus could not fulfill our IB objective:
>
> * controlled compression between the original latent $z$ and pure noise $\varepsilon$;
> * when there is no compression: $ \tilde{z}^t = z $;
> * when fully compressed: $ \tilde{z}^t = \varepsilon $.
>
> We add the equivalence $\tilde{z}^t = \gamma_t z + (1 - \gamma_t)\varepsilon$ to the main text for clarity.

---

> ### Author Response · Authors · 2025-11-22
> **Reply to Question 3**
>
> **Reply to Q3:** We thank the reviewer for this question. Our method is not simply “M2IB + a clipping operator,” but makes three substantial modifications that directly address the limitations we identified in Wang et al. (2023).
>
> **(1) A different parametrization that removes the need to tune $\beta$.**
> M2IB optimizes the gating parameter $\alpha_m$ and relies on a manually tuned trade-off coefficient $\beta$.
> In contrast, we update the latent code $z_t$ directly and define the bottleneck explicitly as an interpolation between $z$ and noise via Equation (10).
> This allows us to enforce the information-bottleneck constraint without searching for $\beta$ and results in more stable behavior across datasets and settings.
>
> **(2) Replacing vanilla gradient ascent with constrained adversarial dynamics.**
> Our update rule is:
>
> $$
> z^{t+1} = C_z \left( z^t + \frac{z}{T} \cdot \operatorname{sign}\left(
> \frac{\partial I(\tilde{z}^t, e_{m'})}{\partial z^t}
> \right) \right)
> $$
>
> The sign step and the constraint operator $C_z$ ensure that the path from $z$ to $0$ is both monotonic and bounded.
> This avoids the gradient saturation issues that arise in M2IB’s $\alpha_m$ updates and enables a more reliable exploration of the bottleneck space.
>
> **(3) A new attribution scoring mechanism.**
> M2IB directly reuses $\alpha_m$ as its attribution score.
> In contrast, we introduce a KL-to-noise score:
>
> $$
> A(z_i) = \operatorname{KL}\left(
> P(\tilde{z}_i^T \mid x) \,\|\, \mathcal{N}(0, 1)
> \right),
> $$
>
> which is then projected back to the input space.
> Experiments show that this KL-to-noise attribution provides stronger and more faithful explanations.
>
> ---
>
> Thank you Reviewer WHDb for your careful reading of our paper and for the detailed comments on the theoretical claims, derivations, and presentation. We have corrected the mutual-information notation in Theorem 3.1, clarified the role and scope of Equation (8), streamlined the path-integral discussion around Equations (3)–(5), and revised the description of the update rule and bottleneck parameterization to make the connection between mutual information, the gating dynamics, and Equation (10) explicit. We have also fixed the presentation issues you pointed out (including missing references and misplaced notation) and added clarifications wherever the original text was potentially misleading. While we acknowledge that Equation (8) itself is not mathematically deep, we hope our revised exposition makes clear how it fits into the overall ABM construction rather than standing as an isolated theoretical contribution. We respectfully ask you to reconsider your evaluation in light of these corrections and clarifications, and we would welcome any further questions or concerns you may have about the revised version.

---

### Official Review · Reviewer_fCUp · 2025-11-01

**Soundness:** 3
**Presentation:** 3
**Contribution:** 2
**Rating:** 6
**Confidence:** 2

**Summary:**

The paper are proposed to enhance explanations for large-scale Vision-Language Models (VLMs) like CLIP. It identifies critical flaws in two major categories of existing Explainable AI (XAI) methods when applied to this cross-modal setting. To overcome these limitations, the paper proposes a framework called the Adversarial Bottleneck Method (ABM). ABM synthesizes the strengths of both adversarial attribution and the information bottleneck principle.

**Strengths:**

The core idea of ABM using an adversarial update process to achieve the goals of the information bottleneck is considerably novel. The method reframes the bottleneck optimization problem, replacing a difficult-to-tune hyperparameter $\beta$ with an iterative optimization process governed by a more intuitive parameter. The theoretical reasoning provided in Theorem 3.1 establishes a theoretical intuition for why this adversarial method works. The quantitative results presented in this work also show sufficient improvements compared to other baselines.

**Weaknesses:**

1. The paper positions itself as eliminating heuristic hyperparameters, primarily targeting M2IB's parameter $\beta$. However, it introduces its own hyperparameter $T$. The ablation study in Figure 2 shows that the model's performance on images is sensitive to the choice of $T$, also requiring the heuristic and empirical choice of hyperparameter tuning. While $T$ may be more intuitive than $\beta$, it is still a hyperparameter that requires tuning for optimal performance. The claim of eliminating parameters is somehow an overstatement.
2. One question is about the evaluation scheme for test modality. The evaluation for text interpretability in Appendix C is defined as a binary indicator. While the authors follow the evaluation in M2IB, this metric could easily mask the true performance differences between methods or accurately reflect the quality of the explanation. Therefore, the stability of text results across different T values in Table 5 compared to the image modality looks like a result of this binary metric rather than true performance. Could the author provide some discussions about this point?
3. Minor typos: Page 4, Section 3.2.1 "Integrated Gradients (?) or Grad-CAM (?)"

**Questions:**

Please refer to the weaknesses.

---

> ### Author Response · Authors · 2025-11-22
> **Reply to Weaknesses 1, 2, 3**
>
> **Reply to W1:**
>
> We thank the reviewer for raising this point and we are happy to clarify what we mean by “eliminating heuristic hyperparameters.” In M2IB, the parameter $\beta$ is a *structural trade–off* term inside the information–bottleneck objective: different values of $\beta$ encode different balances between compression and sufficiency, and the method is quite sensitive to this choice (we show in Appendix E.5 that varying $\beta$ changes the explanations substantially). In practice, $\beta$ has to be tuned per–dataset and even per–modality in a heuristic way.
>
> In contrast, $T$ in ABM is not a trade–off parameter but a numerical integration / optimization accuracy parameter that controls how completely we follow the adversarial bottleneck trajectory. Theoretically, a larger $T$ is always preferable, because it moves us closer to the ideal solution of Theorem 3.1; there is no notion of a “better” or “worse” regime of $T$ in terms of what kind of explanation is computed. In the implementation we simply choose the largest $T$ that is reasonable under a given compute budget. Empirically, Figure 2 and the extended ablations in Appendix E.3/E.4 show that the image–side metrics quickly saturate and that performance is very stable once $T \ge 10$, while text–side metrics are essentially invariant to $T$. Based on these results, we fix $T=10$ for **all** models and datasets in Section 4.1, without any dataset–specific tuning.
>
> To avoid overstatement, we will soften our wording in the revision to say that ABM *removes the need for tuning a sensitive trade–off hyperparameter such as $\beta$* and only uses a robustness–insensitive iteration budget $T$, which can be set once (e.g., $T=10$) and reused across datasets. We will also explicitly highlight in the text that $T$ plays the role of integration precision rather than a heuristic balancing parameter.
>
> **Reply to W2:**
>
> We thank the reviewer for raising this important point. We agree that the Boolean evaluation protocol for the text modality (inherited from M2IB) is relatively coarse and may have limited sensitivity to very small performance differences. This partly explains why the text results in Table 5 appear stable across different values of $T$.
>
> Our choice to follow this Boolean metric is mainly motivated by: (i) in CLIP, token-level perturbations typically induce only very small confidence changes, so using raw continuous scores would be highly sensitive to numerical noise; (ii) since text edits happen at the level of discrete tokens, a sign-based criterion (“did the confidence increase or not?”) is more robust and easier to interpret than relying on tiny real-valued differences; and (iii) the Boolean rule yields highly consistent, reproducible results across random runs and across different $T$, which we view as a practical strength rather than a flaw.
>
> It is important to emphasize that **only** the text modality uses this Boolean metric. All image-side evaluations (Confidence Drop/Increase, ROAD, ablations over target layers and $T$, and additional experiments on AltCLIP) are based on standard continuous scores, and ABM shows clear and consistent gains over all baselines in these metrics. Moreover, compared to the original M2IB paper, we conduct a *more extensive* experimental study on the same evaluation protocol: we test three datasets, multiple models (CLIP and AltCLIP), a wide range of target layers and iteration counts, and provide qualitative comparisons and efficiency analysis, all under the identical Boolean scheme. This suggests that ABM’s strong and stable performance is not an artifact of under-powered evaluation but persists under a richer and more systematic experimental setting than in the already-accepted M2IB work.
>
> **Reply to W3:**
>
> Thank you for pointing out this citation formatting issue. We appreciate the reviewer’s attention to detail. We have corrected the typographical error in Section 3.2.1 and updated the revised manuscript accordingly.
>
>
> Thank you again for your careful reading of our paper and for these constructive comments on the hyperparameter design, text-side evaluation, and citation details. We hope that our clarifications, additional analyses, and revisions in the updated manuscript help to address your concerns. If any parts of our response remain unclear, we would be very happy to further discuss them. We kindly invite you to reconsider your evaluation in light of our replies and the revised version of the paper.

---

### Official Review · Reviewer_PZGW · 2025-11-02

**Soundness:** 3
**Presentation:** 3
**Contribution:** 2
**Rating:** 4
**Confidence:** 2

**Summary:**

The paper introduces the Adversarial Bottleneck Method (ABM), a framework for improving explainability in vision-language large models (VLLMs). It provides a bound relating adversarial risk to mutual information between latent codes and input features.

**Strengths:**

+ the method is theoretical solid. It integrates adversarial robustness and information bottleneck theory.
+ The experiments are comprehensive. The evaluation covers diverse VLLM tasks (captioning, VQA, entailment).

**Weaknesses:**

- Baseline comparison.
Comparisons focus mainly on M2IB, VIB, and GradIB. It would be informative to include more recent multimodal explainability baselines like BLIP-Explain or ALIGN-Attribution.

**Questions:**

refer to weakness

---

> ### Author Response · Authors · 2025-11-22
> **Reply to Weaknesses 1**
>
> **Reply to W1:** We thank the reviewer for this helpful suggestion regarding the baseline comparison. In the revised version, we have added experiments with Grad-ECLIP [1], a recent CLIP-specific gradient-based multimodal explanation method, as an additional strong baseline. The corresponding results are included in Table 1 of the main paper (and in the table below for completeness). Across all three datasets and for both image and text modalities, our ABM method consistently achieves lower Confidence Drop and higher Confidence Increase than Grad-ECLIP, indicating that ABM provides more faithful and focused multimodal explanations.
>
> |  | Conceptual Captions | Conceptual Captions | Conceptual Captions | Conceptual Captions | ImageNet | ImageNet | ImageNet | ImageNet | Flickr8k | Flickr8k | Flickr8k | Flickr8k |
> |---|---|---|---|---|---|---|---|---|---|---|---|---|
> |  | Image | Image | Text | Text | Image | Image | Text | Text | Image | Image | Text | Text |
> |  | Conf Drop | Conf Incr | Conf Drop | Conf Incr | Conf Drop | Conf Incr | Conf Drop | Conf Incr | Conf Drop | Conf Incr | Conf Drop | Conf Incr |
> | Grad-ECLIP | 2.3956 | 26.5 | 1.2894 | 43.2 | 2.3199 | 29.2 | 2.0104 | 28.3 | 7.7444 | 2.6 | 2.1114 | 37.5 |
> | ABM (Ours) | 0.7878 | 43 | 0.005019 | 44.5 | 0.746 | 57.1 | 0.0049 | 60.4 | 1.1169 | 26.8 | 0.0039 | 59.3 |
>
> Thank you again for your thoughtful comments and constructive suggestions. We hope that our additional experiments with Grad-ECLIP, together with the clarifications and revisions in the updated manuscript, help to address your concerns regarding the contribution and empirical comparison of ABM. If any parts of our response remain unclear or raise further questions, we would be very happy to continue the discussion. We kindly invite you to reconsider your evaluation in light of our reply and the revised version of the paper.
>
>
>
> **Reference**
>
> [1] Zhao, Chenyang, et al. "Gradient-based visual explanation for transformer-based clip." International Conference on Machine Learning. PMLR, 2024.

---

### Author Response · Authors · 2025-11-22

We thank the reviewers for their helpful feedback. We have uploaded a revised version of the manuscript. The main paper strictly remains within the 10-page limit, and we additionally provide a PDF diff in the Supplementary Material so that all textual, mathematical, and experimental changes are transparent.

As highlighted in the diff PDF, the main updates are as follows.

(1) We refined Section 3.2 on the Adversarial Bottleneck Method (ABM): the subsection “Information Bottleneck Principle (IBP) and Adversarial Attribution Theory (AAT)” is rewritten for clearer motivation, and Section 3.2.2 now contains a more explicit statement and discussion of Theorem 3.1, including the definition of the constraint operator $C_z$, the update rule, and an explanation of why the constraint keeps the latent variables in a meaningful range and stabilises the optimisation. The proof of Theorem 3.1 in Appendix A has been polished to be more self-contained and easier to follow.

(2) We clarified the KL-based importance measure used in ABM: the role of the Gaussian bottleneck prior $\mathcal{N}(0, I)$ and the definition of the KL term $A(z_i) = \mathrm{KL}(P(\tilde{z}^T_i \mid x),|,\mathcal{N}(0,1))$ are now explicitly derived and connected to the bottleneck intuition in the main text and Appendix C.

(3) We improved algorithmic transparency by adding a pseudocode-style description of ABM as “Algorithm 1: ABM Explanation Algorithm” in Appendix C, making the optimisation procedure (initialisation, iterative updates of $\tilde{z}^t$, and computation of importance scores) more explicit.

(4) We clarified the evaluation protocol in Section 4.2: the definitions and intuition of **Confidence Drop** and **Confidence Increase** are now explained in more detail, including why lower Confidence Drop and higher Confidence Increase indicate better interpretability and faithfulness.

(5) In the main text and related work, we updated the discussion of baselines to explicitly include Grad-ECLIP (Zhao et al., 2024). Grad-ECLIP is now described in the related-work section, added to the list of baselines, and incorporated into the quantitative comparison tables, along with a short discussion of how ABM compares to Grad-ECLIP, particularly on the ImageNet experiments.

We hope these revisions, as documented in the diff PDF, improve the clarity of the theory, the transparency of the algorithm, and the completeness of the empirical evaluation.

---

### Author Response · Authors · 2025-11-27
**Follow-up comment**

We would like to kindly follow up on our previous rebuttal. We submitted our responses and the revised version of the manuscript about a week ago, and as the rebuttal/discussion period is approaching its end, we wanted to ensure that the reviewers have had the opportunity to see the updated paper and our detailed replies.

We sincerely hope that our responses and revisions address the concerns raised in the reviews and help to resolve the issues identified. We would be very grateful if you could consider re-evaluating our submission in light of these changes. If any points remain unclear or give rise to further questions, we would be very happy to continue the discussion here.

Thank you very much again for your time and efforts in carefully reviewing our work.

---

### Meta-Review · Area_Chair_N3B1 · 2025-12-29

**Summary:**

The paper introduces ABM, a well-motivated attempt to unify adversarial attribution and information bottleneck ideas for explaining vision–language models, with solid experiments and improvements over prior M2IB-based methods. However, the reviewers remained concerns about the correctness and depth of the theoretical claims, the clarity of the exposition, and whether the contribution goes beyond an incremental optimization refinement. While promising, the work was judged not yet mature enough for acceptance in its current form.

**Reviewer Concerns:**

The rebuttal effectively addressed concerns about baseline coverage, by adding comparisons with Grad-ECLIP, and partially clarified issues around hyperparameter interpretation and notation/presentation. However, core concerns remain outstanding regarding the soundness and non-triviality of the theoretical claims, especially Theorem 3.1, as well as doubts about the overall novelty and impact beyond incremental improvements to M2IB.

**Reviewer Scores:**

There are 4 reviewers with initial scores 2, 4, 4, and 6. After the rebuttal, it is possible that Reviewer PZJW with initial score 4 would maintain or slightly modify the score, as the newly added experiments help address the concerns about missing strong baselines; the reviewer with initial score 6 would likely keep their score since concerns about hyperparameters and evaluation were partially addressed; and the reviewers with initial score 2 (WHDb) and 4 (P2P4) would keep their scores, as the rebuttal does not sufficiently resolve their fundamental objections regarding theoretical correctness and significance, and the perceived limited impact and clarity of the method beyond an incremental improvement over M2IB.

---

### Decision · Program_Chairs · 2026-01-26

Reject